# COVID RADAR app: Description and validation of population surveillance of symptoms and behavior in relation to COVID-19

**Willian J. van Dijk**[1], **Nicholas H. Saadah**[1,2], **Mattijs E. Numans**[1], **Jiska J. Aardoom**[1,2], **Tobias N. Bonten**[1,2], **Menno Brandjes**[3], **Michelle Brust**[1], **Saskia le Cessie**[4,5], **Niels H. Chavannes**[1,2], **Rutger A. Middelburg**[1,4], **Frits Rosendaal**[4], **Leo G. Visser**[6], **Jessica Kiefte-de Jong**[1,2] *

1 Department of Public Health & Primary Care/LUMC Campus, The Hague Leiden University Medical Center, The Hague, Netherlands, 2 National eHealth Living Lab (NeLL), Leiden University Medical Center, Leiden, The Netherlands, 3 LogiqCare, Ortec B.V., Zoetermeer, The Netherlands, 4 Department of Clinical Epidemiology, Leiden University Medical Center, Leiden, The Netherlands, 5 Department of Biomedical Data Sciences, Leiden University Medical Center, Leiden, The Netherlands, 6 Department of Infectious Diseases, Leiden University Medical Center, Leiden, The Netherlands

* J.C.Kiefte@lumc.nl

**Data Availability Statement:** Because of legal restrictions in both the agreement between app users and the LUMC (research institution) and in

## Abstract

### Background

Monitoring of symptoms and behavior may enable prediction of emerging COVID-19 hot-spots. The COVID Radar smartphone app, active in the Netherlands, allows users to self-report symptoms, social distancing behaviors, and COVID-19 status daily. The objective of this study is to describe the validation of the COVID Radar.

### Methods

COVID Radar users are asked to complete a daily questionnaire consisting of 20 questions assessing their symptoms, social distancing behavior, and COVID-19 status. We describe the internal and external validation of symptoms, behavior, and both user-reported COVID-19 status and state-reported COVID-19 case numbers.

### Results

Since April 2nd, 2020, over 6 million observations from over 250,000 users have been collected using the COVID Radar app. Almost 2,000 users reported having tested positive for SARS-CoV-2. Amongst users testing positive for SARS-CoV-2, the proportion of observations reporting symptoms was higher than that of the cohort as a whole in the week prior to a positive SARS-CoV-2 test. Likewise, users who tested positive for SARS-CoV-2 showed above average risk social-distancing behavior. Per-capita user-reported SARS-CoV-2 positive tests closely matched government-reported per-capita case counts in provinces with high user engagement.

the European General Data Protection Regulation, the authors are not allowed to share the entire dataset online. The data contain potentially identifying information (location, age and gender) and sensitive patient information (for example SARS-CoV-2 test result). Aggregated data can be shared upon request with "medical government or academic research institutions" only. These data requests can be sent to the dedicated data manager I. De Jong (I.de_Jonge@lumc.nl) of the METC of Leiden University Medical Center (metc-ldd@lumc.nl).

**Funding:** Funding for the project was obtained from ZonMW (the Netherlands Organization of Health Research and Development, grant numbers: 10430042010016, 10430022010001 and 10430032010011). The company, ORTEC, provided support in the form of the salary of author MeB. The funder provided support in the form of salaries for authors WJD and NS. The funders had no role in study design, data collection and analysis, decision to publish, or preparation of the manuscript.

**Competing interests:** Salaries of first and second authors were funded by the ZonMW grants. ORTEC provided support in the form of the salary of author MeB. The other authors have declared that no competing interests exist. This does not alter our adherence to PLOS ONE policies on sharing data and materials. The restrictions on sharing of data are due to national legal regulations.

## Discussion

The COVID Radar app allows voluntarily self-reporting of COVID-19 related symptoms and social distancing behaviors. Symptoms and risk behavior increase prior to a positive SARS-CoV-2 test, and user-reported case counts match closely with nationally-reported case counts in regions with high user engagement. These results suggest the COVID Radar may be a valid instrument for future surveillance and potential predictive analytics to identify emerging hotspots.

## Introduction

The world is in the throes of the coronavirus-disease-2019 (COVID-19) pandemic with more than 100 million cases and over 2 million confirmed deaths worldwide as of December 2020 [1]. In the Netherlands, the first case of COVID-19 was diagnosed in February 2020 and since then over one million cases and 17,500 deaths have been confirmed [2]. To date more than 60,000 COVID-19 patients have been admitted to Dutch hospitals, with over 12,000 of these eventually admitted to intensive care [2]–this in a country with just over 1,000 intensive care beds [3]. The strategies of Test Trace and Isolate (TTI), and of measures intended to reduce social contact, have been widely adopted to "flatten the curve" [4, 5]. An important limitation of the TTI strategy is transmission of Severe Acute Respiratory Syndrome Coronavirus 2 (SARS-CoV-2) by COVID-19 carriers without symptoms. Given their lack of symptoms, they may not be tested and remain unidentified by the TTI process despite being a possible source of viral transmission [6]. Recent studies show that this subpopulation may account for as much as half of COVID-19 transmissions [6, 7].

An instrument to continuously monitor social-distancing behavior and symptoms in the population at a local level may support and improve the TTI process by decreasing the delay in identification of risk areas and populations. Research using voluntary symptom self-reporting apps performed in the United Kingdom, the United States of America, and Israel show promising results in the local prediction of COVID-19 using symptom-based tracking [8–10]. However, we find no apps using voluntary social-distancing behavior-reporting to track local COVID-19 hotspots.

During the first COVID-19 wave in the Netherlands, the Leiden University Medical Center (LUMC) and the tech company ORTEC developed and introduced the COVID Radar app. This questionnaire-based app allows individuals to anonymously report COVID-related symptoms and social-distancing behaviors on a regional and population level. The app provides users with direct feedback on, and peer comparison with, their reported social-distancing behavior and symptoms. Our theory is that tracking of symptom and social-distancing behavior data at a population level can be used to identify regions where more COVID-19 cases will subsequently occur, allowing (regional) policy makers and healthcare professionals to affect changes to regulations earlier, and thus more effectively.

In this first descriptive study, our aim is to observe the associations between self-reported symptoms, social-distancing behavior, and self-reported COVID-19 infection by the app's users (i.e. criterion validity), and the associations between these variables and state-reported COVID-19 infections by the National Institute for Public Health and the Environment (i.e. external validation).

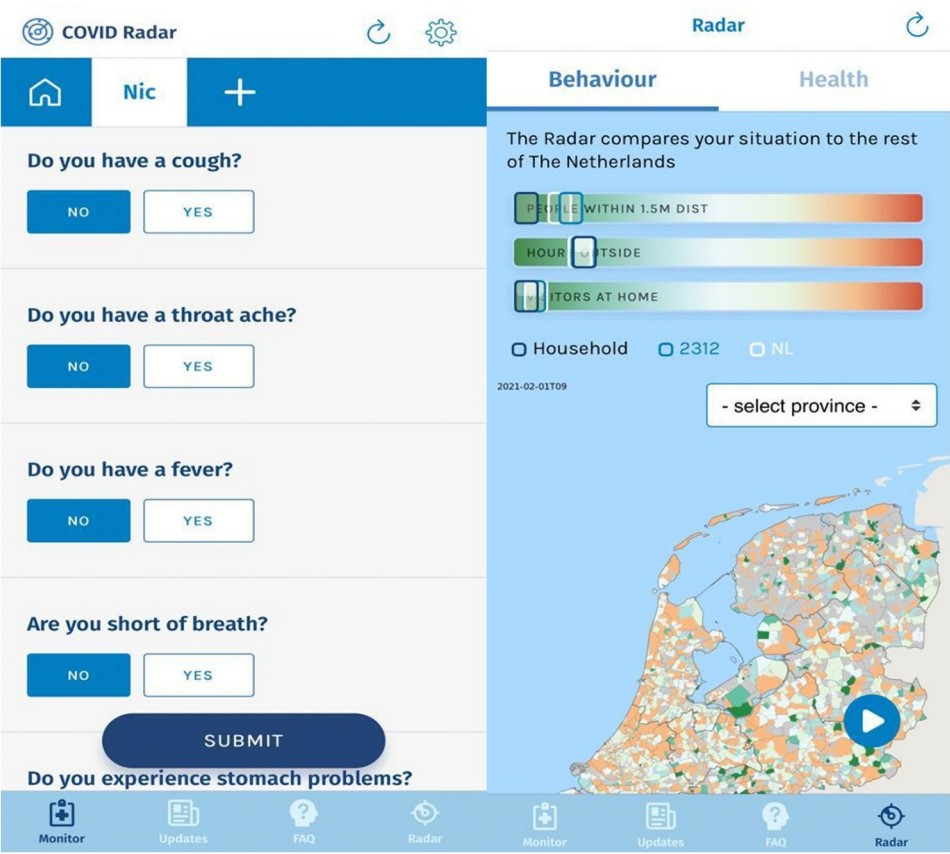

**Fig 1. Screenshot of COVID radar app (Left: Questions. Right: Feedback) "Reprinted from "urlmaps.com" under the Creative Commons Attribution License, with permission from "i-mapping", original copyright 2021".**

## Methods

### COVID radar app

The COVID radar app was released on the 2nd of April 2020 following a short publicity campaign in the local and national media [11]. The app is free to download and allows for multiple user accounts from the same household on one smartphone. The app is not age-limited, meaning children are allowed to download and use the app. Over 85% of the households in COVID radar's user population with minors under 18 years of age are linked to an adult smartphone. Upon first use of the app, users are asked to provide informed consent to share the following information with the research institution as stipulated by the conditions of the European General Data Protection Regulation. Users may opt out by either removing the app or by requesting the data manager to remove all data collected from that individual. Users are asked to register by entering the four digits of their postal code, gender (Male/Female/Other/Not Specified), age category (0–5, 6–11, 12–18, 19–29, ten-year increments from 30–80 and a category for 80+), and occupation (healthcare, education, catering industry, or other occupation with high risk of close contact). Following the initial setup, users are asked to report their symptoms and behavior daily via a questionnaire. A push-reminder is sent every-other day to users reminding them to do so. Fig 1 shows screenshots of the app and S1 Table shows a list of the questions users are asked.

Each observation is comprised of questions assessing symptoms, social-distancing behavior, whether or not the user has been exposed to an individual with COVID-19 in the past 2 weeks, and a user's COVID-19 test history. The questions asked were periodically updated with the addition/removal dates of each question detailed in the online supplement. Via maps displayed within the app, users are presented with regional incidences of symptoms and personal feedback on their social-distancing behavior compared to regional and national means (Fig 1).

Data are transferred daily to a safe data environment within the Information Technology system of the LUMC (S2 Fig). Following importation of the daily data, we exclude observations from users who had requested to opt out, observations listing nonexistent postcodes, and double measurements within one user. Given users are asked if they have tested positive for SARS-CoV-2 within the past two weeks, we considered users SARS-CoV-2 positive/negative if they indicated a SARS-CoV-2 test result at least twice in the app, with the date of the first report used as day zero. More details on the development of the app, selection of the chosen questions, (external) data sources, and data cleaning is available in the supplement. Ethical approval was provided by the Medical Ethical Board of the LUMC (dossier number N20.067), which gave permission to refrain from obtaining consent from parents or guardians as data collection was anonymous. Only data on age category, profession and four digit postal code was collected rendering the data was untraceable to an individual.

## Comparison of included/excluded observations

Following the data cleaning process detailed in the online supplement, we compared the available data in the excluded cohort with that of the included cohort. For each of the binary (symptom) variables collected by the app, we compared the proportion of excluded and included observations reporting this symptom. For each of the continuous social-distancing behavior variables, we compared the mean values for the included/excluded cohorts.

## Descriptive statistics

To describe participant characteristics, we used histograms to explore age distributions of the app users, the number of times the app was used each day, and the number of times individual users used the app. We further compared age, gender, and profession for users ever having tested positive with those never having tested positive for SARS-CoV-2.

## Validation testing

Given the eventual goal of the COVID Radar app is to predict emerging hotspots, we tested the expected associations between symptoms/behavior and SARS-CoV-2 test outcome. We used user-reported test results as our outcome measure for criterion validity testing and cases reported by the National Institute for Public Health and the Environment (RIVM) as our outcome measure for external validation [2].

## Criterion validity

As a measure of criterion validity, we explored associations between the binary symptom variables (e.g., cough, sore throat, loss of smell/taste) and the continuous social-distancing behavior variables (e.g., number of house outside house, number of people within 1.5m) within the cohort of users ever reporting a SARS-CoV-2 test. For users within this ever-tested cohort, we used the date of the test as day 0 and observed the 21 days before and after the test. We calculated the daily mean or proportion for each variable for the entire user- cohort. We then calculated the difference between ever-positive or ever-negative users' reported values and the mean

values for the entire user-cohort on that day. By comparing data from the same days, we eliminated bias introduced by variations in time due to the various lock-down measures implemented during the observation window, as well as seasonal effects on symptoms. The mean values and 95% confidence intervals for these differences were then plotted to show how the ever-positive and ever-negative cohorts compared to the cohort as a whole with regard to these variables in the days surrounding a test. Given the formulation of the question ("Have you tested positive/negative for SARS-CoV-2 in the past two weeks"), the date of the test cannot be determined for those answering this question in the 14 days following the implementation of the question about testing in the app. Given this and the fact that this analysis involved looking at the 14 days prior to a test, users reporting a SARS-CoV-2 test in the 14 days following implementation of the question about testing were not included in this analysis.

### External validation

As a measure of external validation, we compared per-capita user-reported COVID-19 status among the 12 Dutch provinces with per-capita rates as reported by RIVM over the course of the pandemic [2]. Within each province, we plotted 7-day backward looking moving averages of the daily proportion of users reporting each symptom variable alongside the daily nationally reported COVID-19 case counts and the weekly proportions of users reporting each symptom variable alongside the number of Rhinovirus cultures reported by Dutch laboratories [12]. We further plotted daily means and 7-day backward looking moving averages of each social-distancing behavior variables and qualitatively observed how well they reflect nationally applied lockdown-measures and holidays.

### Sensitivity analyses

We repeated the above-described analyses for (a) the cohort of users using the app an above-median number of times during the observation period, (b) the cohort excluding healthcare professionals, and (c) the cohort excluding inhabitants of the province 'Zuid-Holland', the home province of the LUMC where the app was created and users were most exposed to COVID Radar app-related media and advertisements. All statistical analyses were performed in STATA 16.1 (StataCorp, College Station, USA). STATA syntaxes for all analyses are provided in the online supplement.

## Results

In the period 2 April, 2020 to 31 January, 2021 (305 days), the COVID Radar app was downloaded by 278,523 unique users who filled in the in-app questionnaire 6,202,606 times. A total of 102,445 (1.65%) observations were excluded (S4 Fig).

### Comparison of included/excluded observations

The data for the 102,445 (1.65%) excluded observations were fairly representative of the included observations' data in terms of symptoms and behavior. However, excluded observations were less often from a health professional and showed a slightly different age distribution (i.e. older age groups are over-represented in the excluded cohort) (see S2 Table).

### Descriptive statistics

The age distribution of the app's users showed a fairly consistent distribution of users 18–69 years old, and an under-representation of young (<18) and old (>70) users. Female users were overrepresented compared to national figures (See S5 Fig). The number of observations

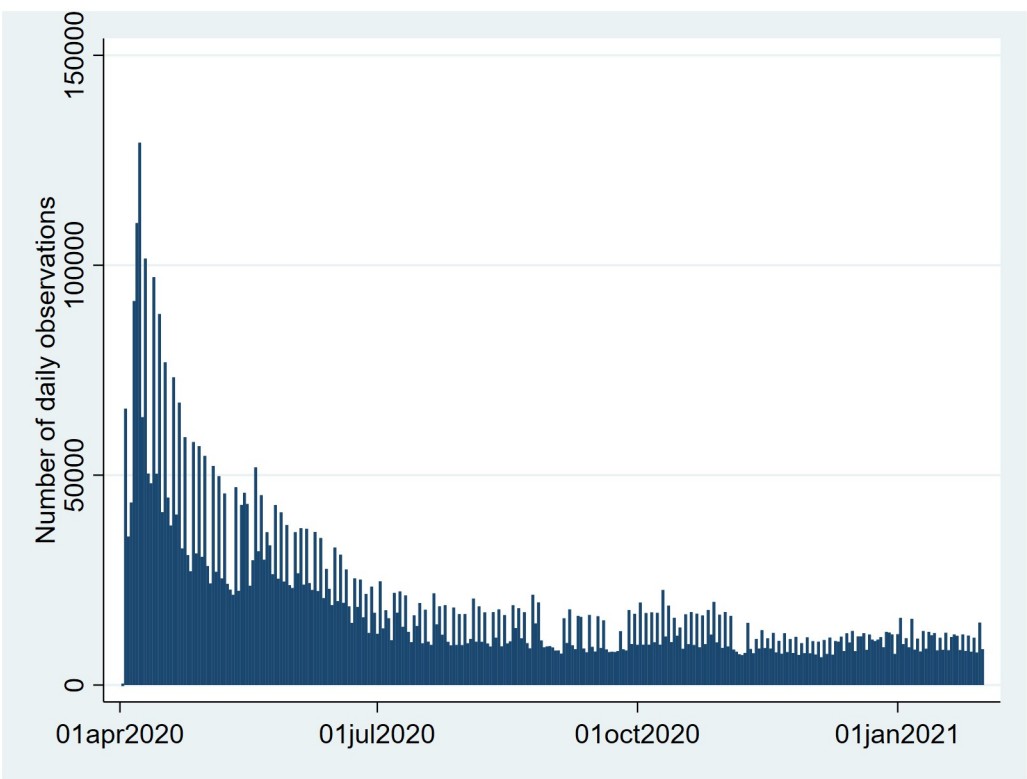

**Fig 2. Number of observations per day.**

(questionnaires answered) per day dropped from over 100,000 in the first week of the app to a steady-state of around 10,000 observations per day during the course of the observation window (2 April, 2020 to 31 January, 2021) (See Fig 2).

The effects of the push reminder sent every-other day to all users is seen in the periodicity in the number of observations between even and odd days. The number of daily observations was highest in the province Zuid-Holland, the home province of the LUMC where the app was conceived and advertised (see Fig 3).

The number of observations per user ranged from 1 to 305 with a median value of 6 ($25^{th},75^{th}$ percentiles [$p_{25}, p_{75}$] = 2, 21) (See Fig 4).

## Criterion validation

From a total of 278,523 unique users, 1,981 (0.71%) reported ever testing positive and 1214 (0.44%) negative for SARS-CoV-2. Ever-positive users were more likely to be women, older than 40 years of age, and healthcare professionals (Table 1).

The proportion of users reporting the eight symptom variables increased beginning approximately 7 days prior to a positive test. This increase was smaller in the cohort of negative tested users (Fig 5a and 5b).

The continuous social-distancing behavior-based variables likewise showed above-mean values in this ever-positive cohort until approximately 7 days prior to a positive test, at which point they sharply decreased to remain below-mean in the week before and after a positive test. These fluctuations were not seen in users testing negative for SARS-CoV-2 (see Fig 6a and 6b).

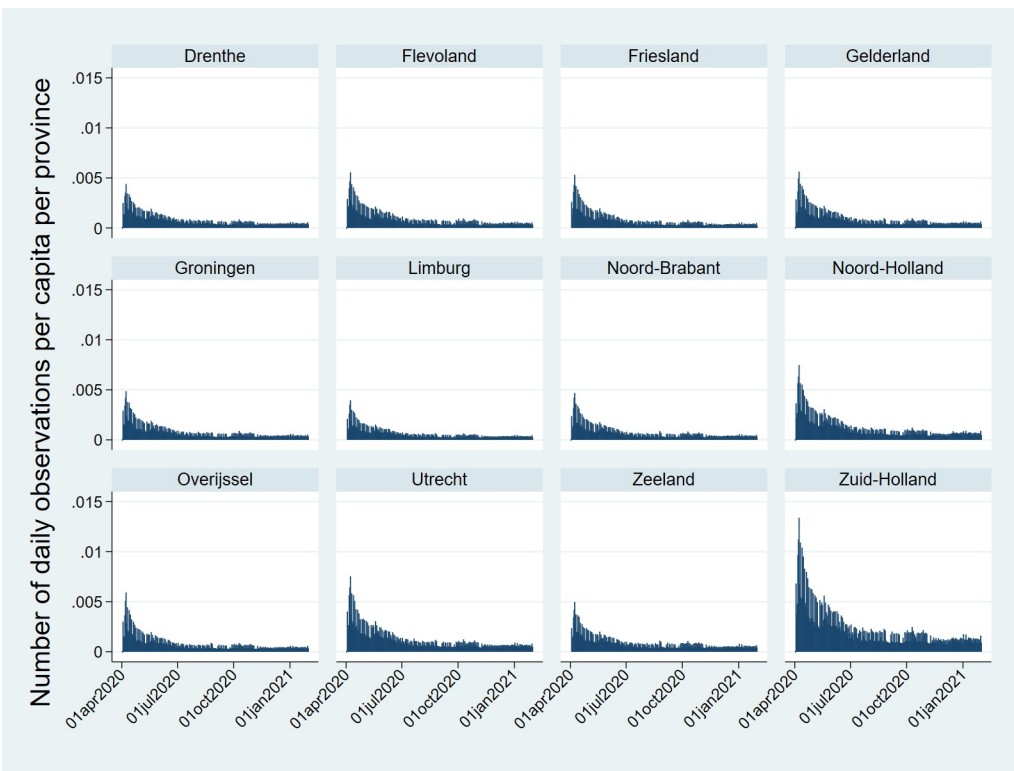

**Fig 3. Number of observations per capita per province in the Netherlands.**

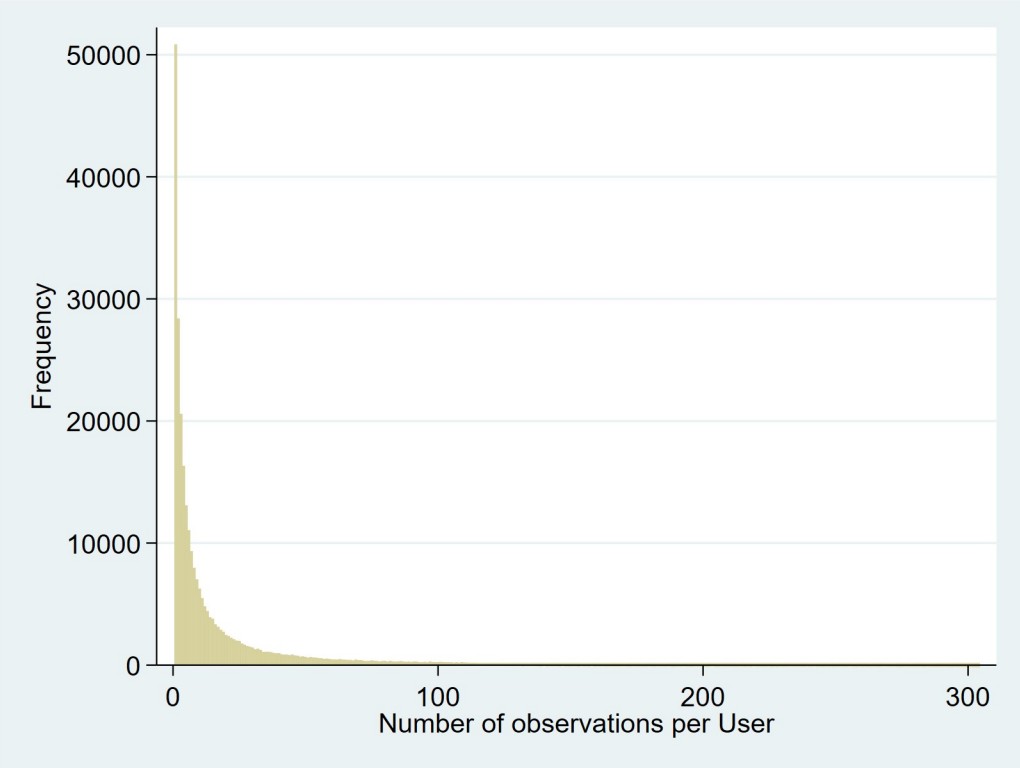

**Fig 4. Distribution of number of observations per user.**

**Table 1. Comparison of users never vs. ever tested SARS-CoV-2 positive.**

| | | | | |
|---|---|---|---|---|
| Number of days | 305 | | | |
| Number of users | 278.523 | | | |
| Number of observations | 6.100.261 | | | |
| Observations per user, median and Interquartile range (IQR) | 6 | | [2; 21] | |
| | Never SARS-CoV-2 + | | Ever SARS-CoV-2+ | |
| | N | (%) | N | (%) |
| Total | 276.542 | 99,3% | 1.981 | 0,71% |
| Gender (female) | 152.515 | 55,2% | 1.201 | 60,6% |
| Profession | | | | |
| Catering | 259 | 0,1% | 9 | 0,5% |
| Education | 1.427 | 0,5% | 77 | 3,9% |
| Healthcare | 28.830 | 10,4% | 372 | 18,8% |
| Other | 16.510 | 6,0% | 412 | 20,8% |
| Other_Contact | 1.239 | 0,4% | 35 | 1,8% |
| Missing | 228.277 | 82,5% | 1.076 | 54,3% |
| Age | | | | |
| 0–5 | 9.033 | 3,3% | 1 | 0,1% |
| 06–11 | 11.273 | 4,1% | 17 | 0,9% |
| 12–18 | 17.635 | 6,4% | 72 | 3,6% |
| 19–29 | 34.309 | 12,4% | 102 | 5,1% |
| 30–39 | 39.305 | 14,2% | 157 | 7,9% |
| 40–49 | 41.887 | 15,1% | 327 | 16,5% |
| 50–59 | 54.718 | 19,8% | 604 | 30,5% |
| 60–69 | 45.993 | 16,6% | 476 | 24,0% |
| 70–79 | 20.292 | 7,3% | 208 | 10,5% |
| 80+ | 2.097 | 0,8% | 17 | 0,9% |

SARS-CoV-2: Severe Acute Respiratory Syndrome Coronavirus 2.

### External validation

As of early January 2021, almost one million cases of COVID-19 had been reported in the Netherlands by the National Institute for Public Health and the Environment (RIVM). The RIVM-reported daily case counts varied from 0 to over 13,000 cases per day. Positive SARS-CoV-2 tests reported in the COVID Radar app alongside the case count as reported by the RIVM for each province show that the association between these two is highest in provinces with a higher number of users, especially Zuid-Holland (Fig 7).

Symptoms and social-distancing behavior varied over time, with both showing a clear temporal association with RIVM-reported case counts over time (Figs 8 and 9).

Plotting the RIVM-reported number of reported positive cultures of Rhinovirus alongside our symptom data suggests variables 'fever', 'pain in the chest' and 'loss of smell' are associated with COVID-19 case count while variables 'coughing' and 'sore throat' correlated more closely with Rhinovirus cultures (Fig 10).

The daily mean number of people within 1.5 meters declined sharply around the middle of September, reflecting the national lockdown measures introduced, and showed peaks during national holidays (Fig 11).

The variable 'number of visitors' likewise showed peaks in the period around Christmas and New Year's Eve (Fig 12).

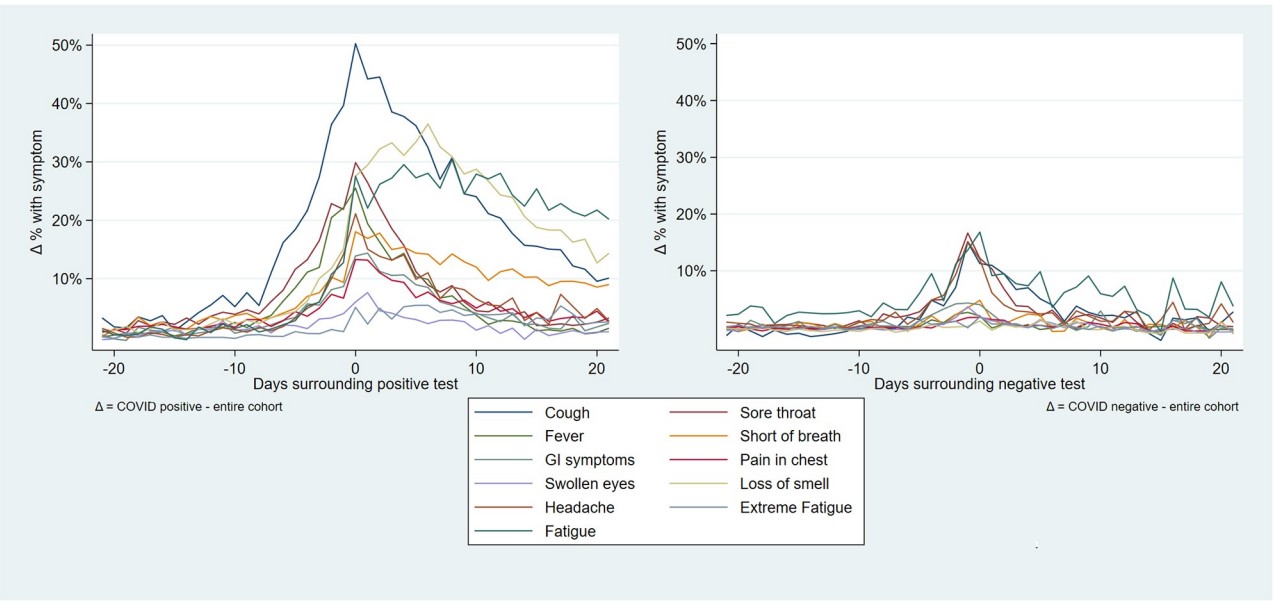

**Fig 5.** a) Daily reported proportion of users reporting symptoms surrounding positive SARS-CoV-2 test. b) Daily reported proportion of users reporting symptoms surrounding negative SARS-CoV-2 test. Data with 95% confidence intervals in online supplement.

## Sensitivity analyses

These analyses were repeated using (a) only users reporting an above median number of observations (referred to as 'faithful' users), (b) only users outside the province Zuid-Holland, and (c) only non-healthcare professionals. Differences in the results for these three sensitivity

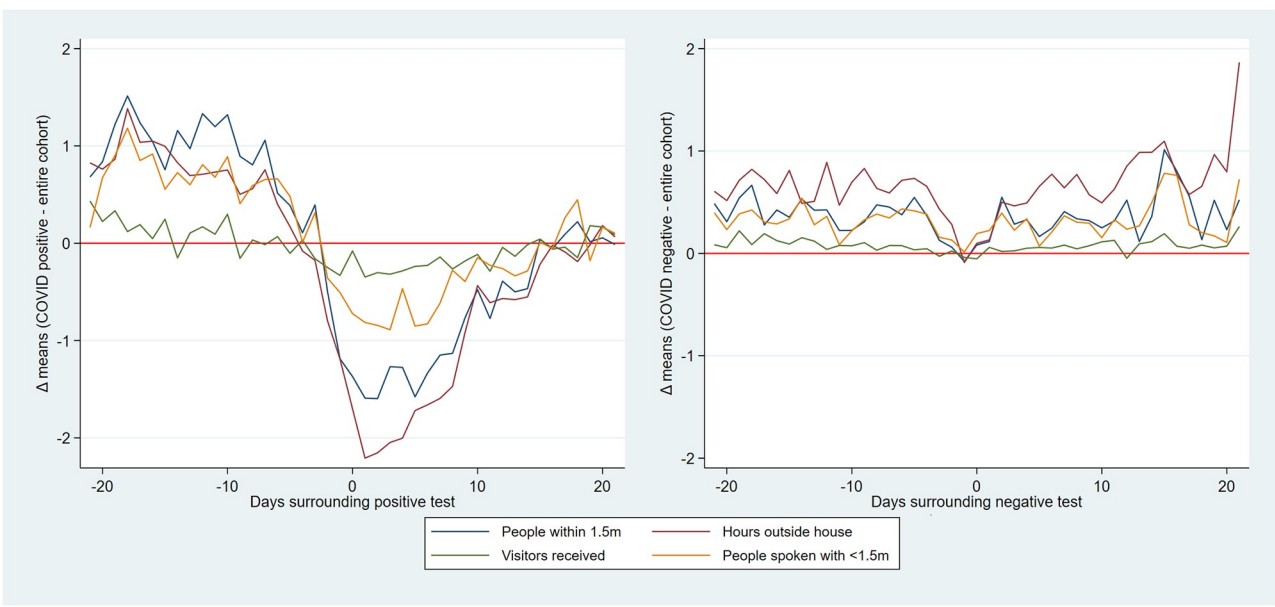

**Fig 6.** a) Mean difference between SARS-CoV-2 positive users and daily means for behavior variables in the days surrounding positive SARS-CoV-2 test. Red line: no difference. (example: on day 20 before test, on average a positive tested user reported one extra person within 1.5m (blue line) compared with the daily mean). B) Mean differences between SARS-CoV-2 negative users and daily means for behavior variables in te days surrounding negative SARS-CoV-2 test. Data with 95% confidence intervals in supplements.

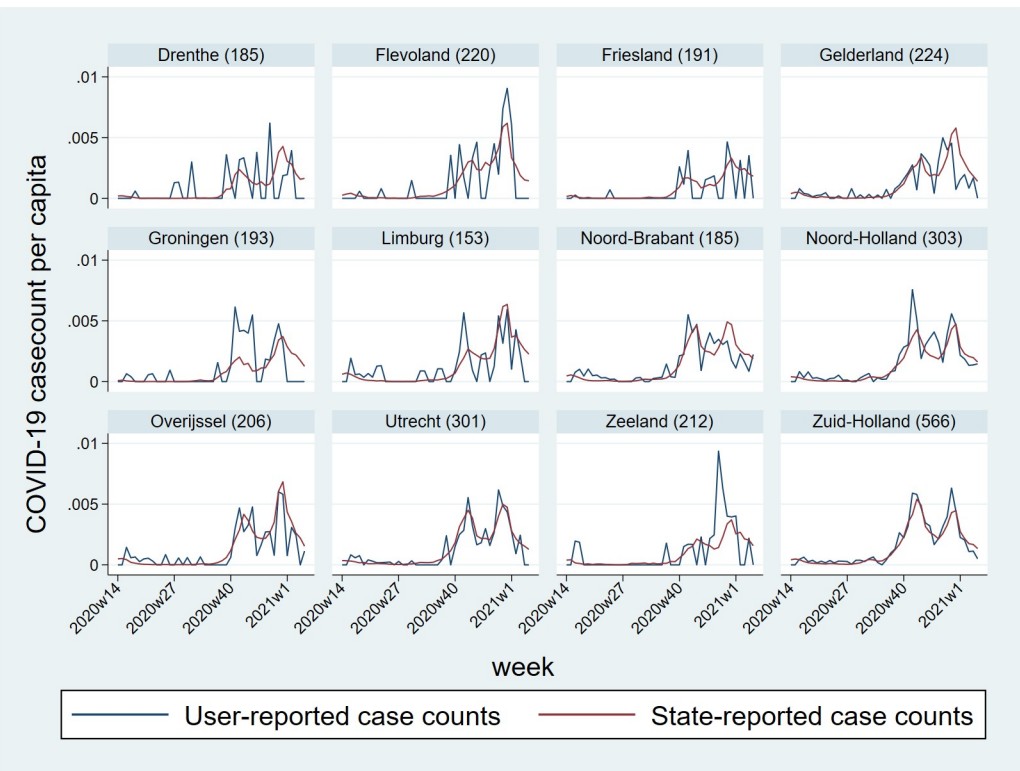

**Fig 7. Weekly proportion of SARS-CoV-2 positive users reported in the COVID Radar app and case count per capita as reported by the RIVM for each province.** (the number written beside the name of each province represents the mean number of weekly unique users per 100.000 inhabitants in that province during the study period).

analyses were minimal and none of the trends seen here were reversed (data shown in supplements).

## Discussion

Since April 2020, the COVID Radar app has collected over 6 million user-provided questionnaires detailing COVID-related symptoms and social-distancing behaviors from over 275,000 unique users within the Netherlands. Symptom and behavior data were temporally associated with user-reported SARS-CoV-2 tests. A correlation between in-app reported case count and national-reported case counts was likewise seen, especially in provinces with high user-engagement. Social-distancing behavior variables showed the expected pattern in relation to national applied lockdown measures and holidays.

### Criterion validity

Our qualitative (visual) association testing showed clear associations between both user-reported symptoms and user-reported social-distancing behavior, and user-reported SARS-CoV-2 test results. While not here quantified, some variables (e.g. 'fever', 'pain in the chest' and 'loss of smell') were more closely associated with case-count than others (e.g. 'coughing' and 'sore throat'), which seemed as associated with Rhinovirus as with SARS-CoV-2. These associations are supported by prior research [13–16]. The pattern of social-distancing behaviors within the cohort of users who eventually report a positive SARS-CoV-2 test was particularly interesting. This cohort showed above-mean risk social-distancing behavior (e.g. more

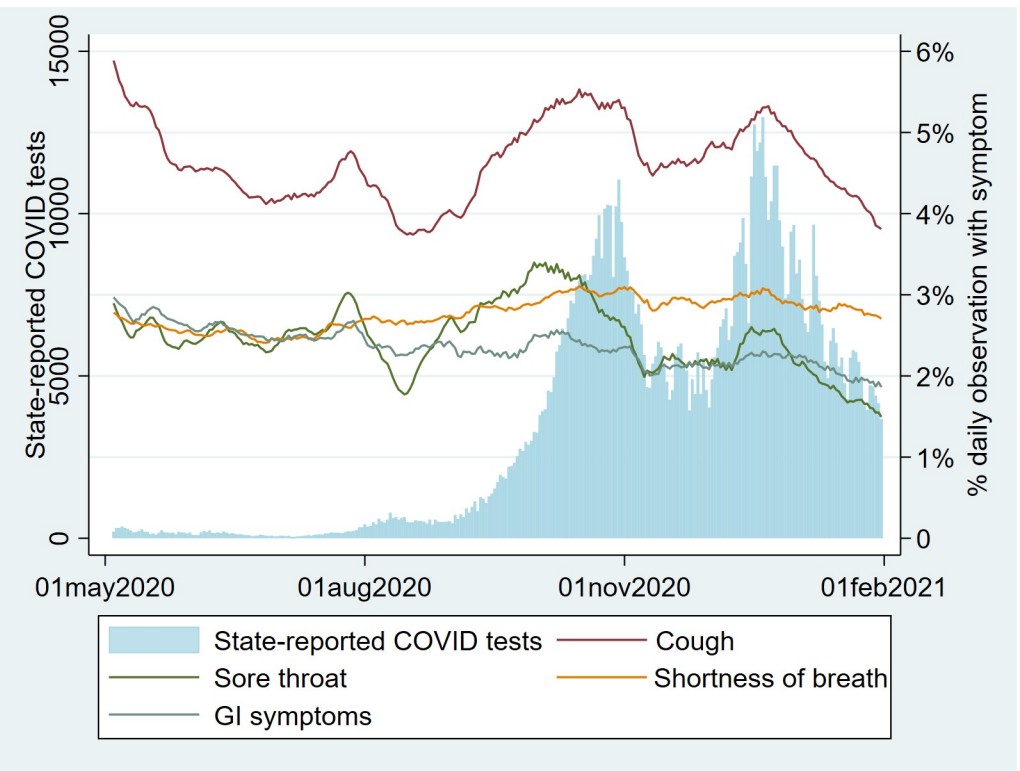

**Fig 8. Symptoms vs. time alongside number of nationally reported positive SARS-CoV-2 tests (percentage of users reporting cough, sore throat, GI (gastrointestinal) symptoms, and shortness of breath).**

people within 1.5m, more visitors at home) between 20 and 10 days prior to a positive test (i.e. the period during which transmission likely occurred), at which point their social-distancing behavior quickly drops to a below-mean value as they became symptomatic and decided to be tested. The extent of above mean risk behavior was lower in users eventually testing negative.

## External validity

Comparing COVID radar data to external data sources showed logical (temporal) associations in symptoms, social-distancing behavior, and test results. The strongest associations were observed in regions with high user-engagement. Given the symptoms tracked by the app are common both to SARS-CoV-2 and other respiratory tract infections, future efforts directed at prediction will need to correct for Rhinovirus and other viruses using viral surveillance data from Dutch laboratories. The extent and types of restrictions imposed on the Dutch population varied during the observation period and their effects were clearly visible in the social-distancing variables reported by users.

Comparison of excluded and included observations showed slight differences in age distribution but relative consistency in other variables. The small size of the excluded cohort minimized the risk of bias being introduced via this exclusion step. There was a large variance in the number of observations per user, with some users answering questionnaires daily while others filled in the app only once during the observation period. While it is reasonable to assume more faithful users may provide more accurate data, sensitivity analyses performed using data from users with an above-median number of app entries show no significant differences as compared to our primary analyses. The lack of a clear difference in the results when

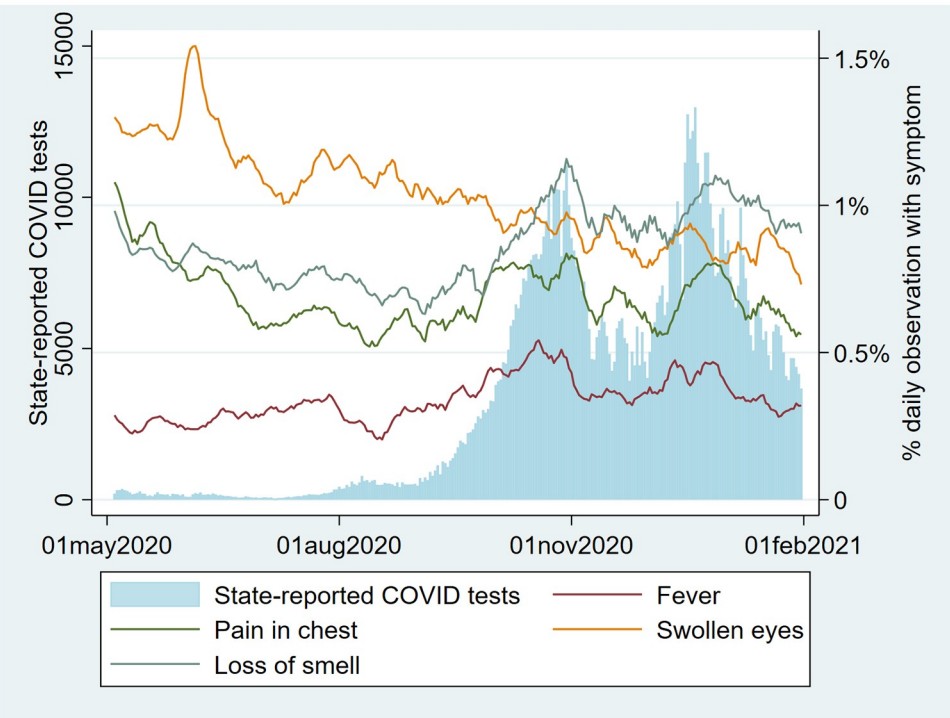

**Fig 9. Symptoms vs. time alongside number of nationally reported positive SARS-CoV-2 tests (percentage of users reporting pain in chest, fever, loss of smell, swollen eyes).**

analyzing users of different engagement-levels suggests any bias introduced by differences in the reporting habits of these users was small.

There was an overrepresentation of users from the province Zuid-Holland in our data, due to Zuid-Holland being the home province of the LUMC, the hospital in charge of app design/analysis. This also likely explains the over-representation of health care professionals, to whom the app was thoroughly advertised within the environment of the LUMC. Despite this overrepresentation, our sensitivity analyses excluding Zuid-Holland users and healthcare professionals showed similar results, suggesting any bias introduced by their overrepresentation is minimal. COVID radar users were more often female and middle-aged. This was due to the overrepresentation of healthcare workers (who were more often female and mid-aged). However the sensitivity analysis excluding healthcare workers resulted in no different conclusions.

Noteworthy too is the fact that fully 30% of those users reporting a positive SARS-CoV-2 test reported no symptoms on the day of the positive test (data shown in S8 Fig). This is in line with the estimated number of COVID-19 carriers without symptoms, as reported by other studies [6, 7]. Our analysis likewise showed loss of smell and cough may continue for weeks following the positive SARS-CoV-2 test, as also confirmed in previous studies [17].

## Limitations

All data in the app was self-reported and thus subjected to differences in personal interpretation of the questions. However, we do not expect differential misclassification as we see logical trends in symptoms and behavior on both individual and national levels. State-reported case counts were those reported by RIVM, whose data should include tests performed in private practices as they are required to be forwarded to RIVM. However, as there is no oversight for

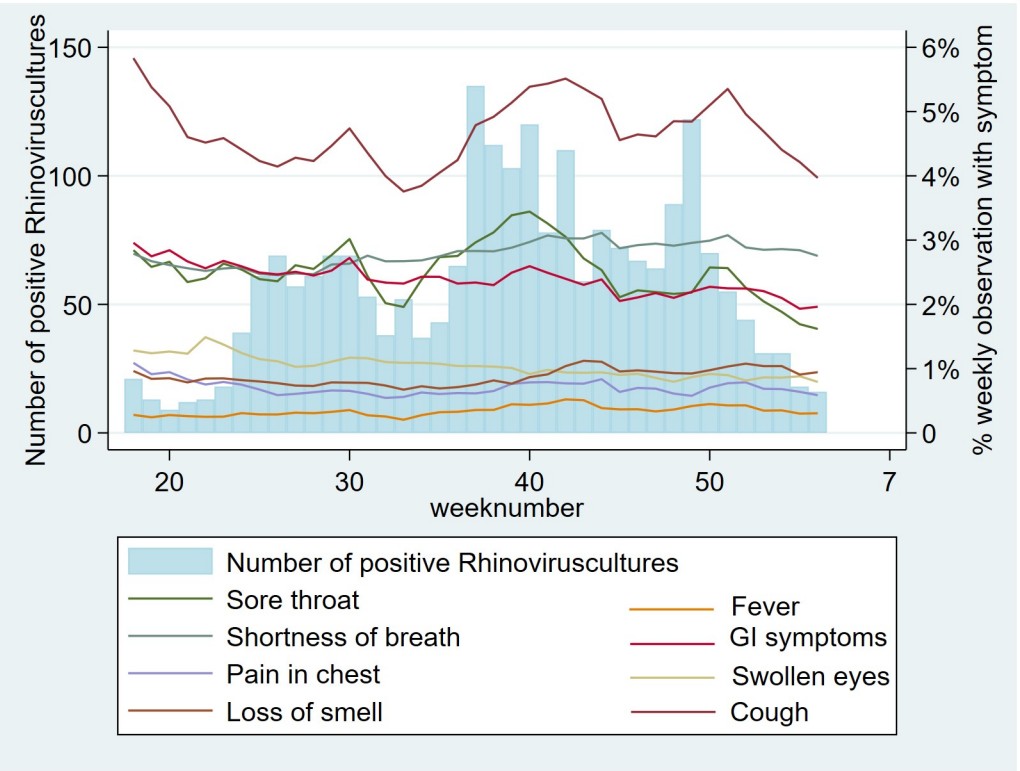

**Fig 10. Symptoms vs. time (weekly) alongside number of nationally reported positive cultures of Rhinovirus (percentage of users reporting symptoms).**

this process, the RIVM-reported case-counts likely represent under-estimates of the number of confirmed cases [18].

COVID radar additionally provided direct feedback to users on how their symptoms and behavior compared to their peers which likely has an effect on user behavior. This may bias the generalizability of COVID radar data, especially behavioral data. The effect of this feedback loop on users' behavior would be expected to lead to an overly conservative estimate of the behavior of the population. Despite this, expected changes in reported behavior in the periods following national holidays and changes to social distancing policies are observed in COVID radar data. Additionally, altered behavior due to app-feedback would be expected to be observed in more loyal users of the app. Our sensitivity analysis on loyal users showed no significant difference in reported behavior. Given these realities, while we accept that app feedback altering user behavior has the potential to bias our results, we feel any bias introduced has been shown here to be small.

Testing capacity in the Netherlands was low during the developmental stage of the app and has increased during the study period. In the final months of 2020, testing was expanded to include those without symptoms. As a result, the prevalence of COVID-19 in the Netherlands could be underestimated. Because of this change in testing policy, the question regarding negative tests was implemented at a later date resulting in less data about negative tests compared to data on positive tests during a shorter period of time. Nonetheless, we were able to show that the association between symptoms and a negative test is less apparent than their associations with a positive test, suggesting our conclusions remain valid. Also, testing of the

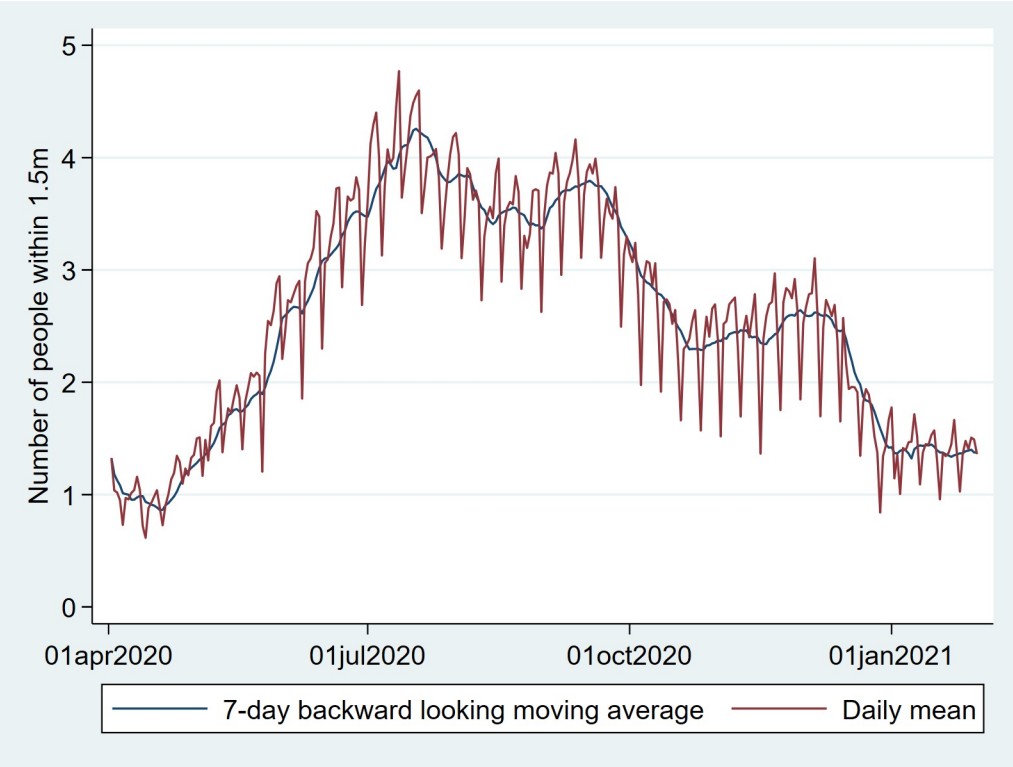

**Fig 11. Daily mean number of people within 1.5 meters (with 7-day backward-looking moving average in black) September: Closing of restaurants.** December: holidays, followed by closing of non-essential shops.

underaged (<12 years) was rare during the study period, resulting in a relatively old SARS-CoV-2 positive cohort in this study.

## Future implications

Having validated the expected associations between symptoms, social-distancing behavior, and COVID case-count, our next steps will involve attempted prediction of emerging hotspots by combining symptom and social-distancing behavior data to quantify risk of COVID-19 cases. Such predictions could be used to help guide COVID-19 policy. Our study indicated the quality of the submitted data is best where user-engagement is high. Prediction-based goals will thus be aided by increasing user count. Regional predictions may additionally be improved through incorporation of data from general practitioners, more detailed demographic data, and mobility data using a machine learning based approach. Another possibility for further research is testing of associations between regional SARS-CoV-2 cases, symptoms, behavior and other regional data related, for example, to the physical environment.

## Conclusion

The COVID Radar app successfully collects anonymous, user-reported data on COVID-19-related symptoms and social-distancing behavior. Initial validation showed symptoms and behavior reported within the app are correlated with in-app reporting of a SARS-CoV-2 test. The predictive potential of the COVID Radar is demonstrated as external validation showed

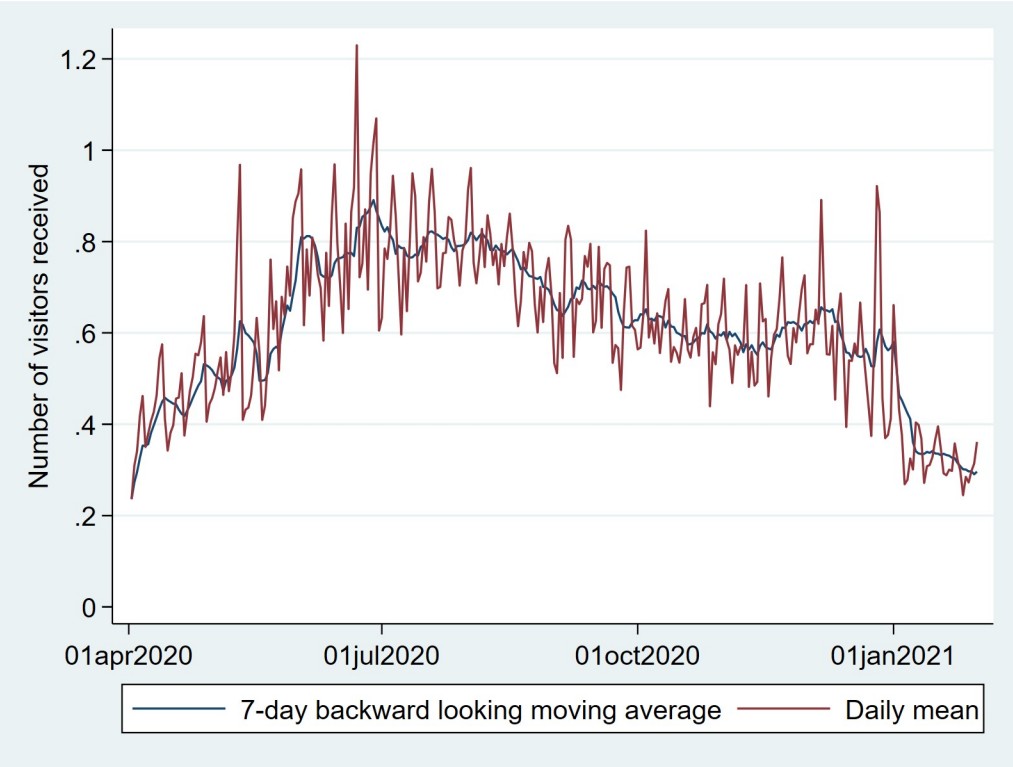

**Fig 12. Daily mean number of visitors vs. time (with 7-day backwards moving average).** Peaks are seen during Christmas and New Year's Eve. The fifth of December is a holiday in the Netherlands known as "Sinterklaasavond".

in-app reported positive SARS-CoV-2 tests track well with state-reported case counts. Future research will focus on regional predictions using these data.

## Supporting information

**S1 Fig. Screenshot of the app.** "Reprinted from "urlmaps.com" under the Creative Commons Attribution License, with permission from "i-mapping", original copyright 2021".
(TIF)

**S2 Fig. Dataflow of data in COVID radar.**
(TIF)

**S3 Fig. Overview of data cleaning.**
(TIF)

**S4 Fig. Flowchart of Inclusion of data.**
(TIF)

**S5 Fig. Age and gender distribution of app's users compared to national data.**
(TIF)

**S6 Fig. Mean difference between values of users reporting symptoms and daily proportion per day surrounding positive SARS-CoV-2 test (red line: No difference with daily proportion) with 95% confidence intervals.**
(TIF)

**S7 Fig. Mean difference between values of users reporting behavior and daily means per day surrounding positive SARS-CoV-2 test (red line: No difference with daily mean) with 95% confidence intervals.**
(TIF)

**S8 Fig. Proportion of users asymptomatic per day surrounding positive SARS-CoV-2 test.**
(TIF)

**S9 Fig. Mean difference between number of symptoms reported by users reporting symptoms and daily mean number per day surrounding positive SARS-CoV-2 test (red line: No difference with daily proportion) with 95% confidence intervals.**
(TIF)

**S10 Fig. As S6 Fig, but than for negative SARS-CoV-2 test.**
(TIF)

**S11 Fig. As. S7 Fig, but than for negative SARS-CoV-2 test.**
(TIF)

**S12 Fig. As S8 Fig, but than for negative SARS-CoV-2 test.**
(TIF)

**S13 Fig. As S9 Fig, but than for negative SARS-CoV-2 test.**
(TIF)

**S14 Fig. Daily mean number of hours out of house (with 7-day backward moving average in black).**
(TIF)

**S15 Fig. Daily proportion of users working/going to school outside of house (with 7-day backward moving average in black).**
(TIF)

**S16 Fig. Daily mean number of persons spoken with within 1.5 meters (with 7-day backward moving average in black).**
(TIF)

**S1 File. Syntax importation data STATA.**
(DO)

**S2 File. Syntax selection of exclusions STATA.**
(DO)

**S3 File. Syntax importation external data STATA.**
(DO)

**S4 File. Syntax cleaning data STATA.**
(DO)

**S5 File. Syntax analyzes data, generation figures and sensitivity analyzes.**
(DO)

**S6 File. Supplemental text overview of development of the app.**
(DOCX)

**S7 File. Sensitivity analyses (S5–S16 Figs repeated without below median reporting users, healthcare professionals and inhabitants of the province Zuid-Holland).**
(ZIP)

**S1 Table. Overview of questions asked in the app.**
(XLSX)

**S2 Table. Comparison of excluded and included data.**
(XLSX)

## Author Contributions

**Conceptualization:** Mattijs E. Numans, Menno Brandjes, Michelle Brust.

**Data curation:** Willian J. van Dijk, Nicholas H. Saadah.

**Formal analysis:** Willian J. van Dijk.

**Methodology:** Willian J. van Dijk, Saskia le Cessie.

**Project administration:** Jessica Kiefte-de Jong.

**Supervision:** Mattijs E. Numans, Jessica Kiefte-de Jong.

**Visualization:** Willian J. van Dijk.

**Writing – original draft:** Willian J. van Dijk.

**Writing – review & editing:** Nicholas H. Saadah, Mattijs E. Numans, Jiska J. Aardoom, Tobias N. Bonten, Menno Brandjes, Michelle Brust, Saskia le Cessie, Niels H. Chavannes, Rutger A. Middelburg, Frits Rosendaal, Leo G. Visser, Jessica Kiefte-de Jong.

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
