## [Decision Letter · Decision Letter 0]

13 Apr 2021

PONE-D-21-03787

COVID RADAR app: Description and Validation of Population surveillance of Symptoms and Behavior in relation to COVID-19.

PLOS ONE

Dear Dr. Kiefte,

Thank you for submitting your manuscript to PLOS ONE. After careful consideration, we feel that it has merit but does not fully meet PLOS ONE’s publication criteria as it currently stands. Therefore, we invite you to submit a revised version of the manuscript that addresses the points raised during the review process.

Please take into consideration all the reviewers comments/issues in editing the new version. Especially pay attention in acknowledging any other potential limitations and biases present in a self-selecting app-based population, and add some more formal analysis to support the claims being made.

We look forward to receiving your revised manuscript.

Kind regards,

Simone Lolli

Academic Editor

PLOS ONE

Journal Requirements:

2. We note that Figure 1 and Figure S1 in your submission contain map images which may be copyrighted.

a. You may seek permission from the original copyright holder of Figure 1 and Figure S1 to publish the content specifically under the CC BY 4.0 license. 

3. Please provide additional details regarding participant consent.

In the ethics statement in the Methods and online submission information, please ensure that you have specified (a) whether consent was suitably informed and (b) what type you obtained (for instance, written or verbal). If your study included minors under age 18, state whether you obtained consent from parents or guardians.

If the need for consent was waived by the ethics committee, please include this information.

'The funding organization had no role in study design, data collection and analysis, decision to publish, or preparation of the manuscript.'

6. Thank you for stating the following in your Competing Interests section: 

'The Covid-Radar was developed by Ortec and is freely available in the app- and playstore. Data of the covid-radar was transferred to LUMC for independent analyses. Funding for the project was obtained from ZoNMW (the Netherlands Organization of Health Research and Development, grant number: 10430042010016, 10430022010001 and 10430032010011. The funders had no role in study design, data collection and analysis, decision to publish, or preparation of the manuscript.'

a. Please complete your Competing Interests statement to state any Competing Interests. If you have no competing interests, please state "The authors have declared that no competing interests exist.", as detailed online in our guide for authors at http://journals.plos.org/plosone/s/submit-now

7. Thank you for stating the following in the Financial Disclosure section:

'The funding organization had no role in study design, data collection and analysis, decision to publish, or preparation of the manuscript.' 

We note that one or more of the authors are employed by a commercial company: Ortec B.V.

Additional Editor Comments:

Please take into consideration all the reviewers comments/issues in editing the new version. Especially pay attention in acknowledging any other potential limitations and biases present in a self-selecting app-based population, and add some more formal analysis to support the claims being made.

Reviewers' comments:

Reviewer's Responses to Questions

**Comments to the Author**

1. Is the manuscript technically sound, and do the data support the conclusions?

Reviewer #1: Yes

Reviewer #2: Yes

2. Has the statistical analysis been performed appropriately and rigorously? 

Reviewer #1: No

Reviewer #2: Yes

3. Have the authors made all data underlying the findings in their manuscript fully available?

Reviewer #1: Yes

Reviewer #2: Yes

4. Is the manuscript presented in an intelligible fashion and written in standard English?

Reviewer #1: Yes

Reviewer #2: Yes

5. Review Comments to the Author

Reviewer #1: This is interesting work. I've a number of commments and suggestions that I think could further improve the quality of this manuscript; including better presentation, more acknowledgement of the potential limitations and biases present in a self-selecting app-based population, and some more formal analysis to support the claims being made.

Abstract, line 63: should be have, not having. And I'm not entirely sure what you mean by "Symptom variables increased noticeably in the week" - do you mean reports of users experiencing symptoms increased?

Were there any limits on who could sign up to the app - e.g., were children allowed to? Were users encouraged to report on behalf of others on the multiple accounts, or were people asked to each fill out their own reports?

Line 126 - I'd prefer 'individual with Covid-19' to the more colloquial 'Corona patient'.

Giving user feedback in the app may alter their behaviour and make the app less representative of the population behaviours. Did you see any evidence of this? I think this merits a bit more dicussion that already present as it has the potentialy to significantly bias your app poulation, and undermine the ability of the Covid RADAR to detect wider population changes.

I think at times the authors have put too much into the supplementary material and it has prevented the manuscript from being self contained. For example, line 135 "Following the data cleaning process detailed in the online supplement..." It may be that the cleaning process is extensive and a full description of it best belongs in a supplement, but I think it would help readers if a brief overviewing of the process was included in the main text.

Many of the figures don't have labels for the y-axis, please add.

There are a huge number of figure. It's good to have included so much detail, but I think some of these could be condensed into single figures with multiple panels. I think this would make it easier for readers to engage with and understand your work; it is difficult to do so when frequently scrolling between the texts and figures. For example, figures 5 and 6 could be combined into one.

It is well documented that populations of participatory digital surveys tend to not be representative of the population - they tend to be more female, younger, more educated. It appears the Covid RADAR is no exception; it would be help if this was made explicitr by comparison of your demographic to the country's general population, and the implications of this discussed. There are also harder to quantify biases: signing up to the app suggests a greater interest in Covid-19 than the average person, which might mean e.g. they are more likely to take the rules seriously and stick to them. It is very difficult to quantify the effect of this but it should be acknowledged as a limitation.

It is suprising that the Covid positive population tended to be older than the negative population. Could you comment on this - does this finding agree with other data sources?

Figure 9: are you able to show number of tests conducted as well as number of positive tests; as it's difficult to see whehther the peaks are true rises in population Covid-19 or just due to increased testing. The comparison is qualitative, and should be supported by measuring the quantiative correleation between your app data and cases; this would also help you formally justify your claim that the correlation is higher in regions with greather app participation.

Your claims that "‘fever’, ‘pain in the chest’ and ‘loss of smell’ are associated with COVID-19 case count while variables ‘coughing’ and ‘sore throat’ correlated more

closely with Rhinovirus cultures" is again, completely based on qualitative comparison and would be strenghtened by quantitative measurements. These findings appear to me consistent with findings in the literature, which is encouraging, and I think this should be noted.

Reviewer #2: The submitted manuscript reports analysis from data obtained from COVID RADAR app available in The Netherlands during COVID-19 pandemic outbreak (2020-2021). Overall, the analysis makes sense and the results are coherent. I have only a minor issue to address. As the app is based on self-reporting by app users, it should be specified that a bias toward serious illness exists, as the hospitalized users into Intensive Care Unit are likely not reporting their symptoms. This is a limitation in the analysis because it would be have been interesting to infer which symptoms and which areas might be prone to serious illness, i.e. also try to assess if pollution is playing a role. Other minor comments regard the manuscript, where sometimes too colloquial words are used, e.g. Corona. Lastly, on plots, x and y axis labels should be added.

6. PLOS authors have the option to publish the peer review history of their article (what does this mean?). If published, this will include your full peer review and any attached files.

Reviewer #1: **Yes: **Mark Graham

Reviewer #2: No

---

## [Author Response · Author response to Decision Letter 0]

3 Jun 2021

Journal Requirements:

We have adjusted the style of the manuscript according to PLOS ONE’s style requirements.

2. We note that Figure 1 and Figure S1 in your submission contain map images which may be copyrighted.

We uploaded the completed Content Permission Form and updated the figure caption.

Lines 124-126 AND 454-455: “Reprinted from “urlmaps.com” under the Creative Commons Attribution License, with permission from “i-mapping”, original copyright 2021.”

3. Please provide additional details regarding participant consent.

In the ethics statement in the Methods and online submission information, please ensure that you have specified (a) whether consent was suitably informed and (b) what type you obtained (for instance, written or verbal). If your study included minors under age 18, state whether you obtained consent from parents or guardians. If the need for consent was waived by the ethics committee, please include this information. 

It is possible for minors under age 18 to download and use the app. Local ethical committee gave permission to refrain from obtaining consent from parents or guardians because of the construction of the data collection as data collection was anonymous: only data on age category and four digit postal code was collected which was untraceable to an individual person individual. 

Methods line 110-115: “Children are allowed to download and use the app. Over 85% of the households with minors under 18 years of age are linked to an adult smartphone. Upon first use of the app, users are asked to provide informed consent to share the following information with the research institution as stipulated by the conditions of the European General Data Protection Regulation. Users may opt out by either removing the app or by requesting the data manager to remove all data collected from that individual.”

Methods line 142-145: “, which gave permission to refrain from obtaining consent from parents or guardians because of the construction of the data collection as data collection was anonymous. As such, only data on age category and four digit postal code was collected which was untraceable to an individual person individual”

4. We note that you have indicated that data from this study are available upon request. 

PLOS only allows data to be available upon request if there are legal or ethical restrictions on sharing data publicly. In your revised cover letter, please address the following prompts:

a. If there are ethical or legal restrictions on sharing a de-identified data set, please explain them in detail (e.g., data contain potentially identifying or sensitive patient information) and who has imposed them (e.g., an ethics committee). Please also provide contact information for a data access committee, ethics committee, or other institutional body to which data requests may be sent.

To the reviewer 

Because of legal restrictions in both the agreement between app users and the LUMC (research institution) and in the European General Data Protection Regulation, we are not allowed to share the entire dataset online. Our data contain potentially identifying information (location, age and gender) and sensitive patient information (for example SARS-CoV-2 test result). Aggregated data can be shared upon request with “medical government or academic research institutions” only. These data requests can be sent to the dedicated data manager I. De Jong (I.de_Jonge@lumc.nl) of the METC of Leiden University Medical Center (metc-ldd@lumc.nl).

'The funding organization had no role in study design, data collection and analysis, decision to publish, or preparation of the manuscript. At this time, please address the following queries:

a. Please clarify the sources of funding (financial or material support) for your study. List the grants or organizations that supported your study, including funding received from your institution.

Funding for the project was obtained from ZonMW (the Netherlands Organization of Health Research and Development, grant numbers: 10430042010016, 10430022010001 and 10430032010011). The company ORTEC provided support in the form of the salary of author MeB. 

b. State what role the funders took in the study. If the funders had no role in your study, please state: 

Salaries of first and second authors were funded by the ZonMW grants. ORTEC provided support in the form of the salary of author MeB. The other authors have declared that no competing interests exist.

6. Please complete your Competing Interests statement 

a. to state any Competing Interests. If you have no competing interests, please state

 "The authors have declared that no competing interests exist.", 

ORTEC provided support in the form of the salary of author MeB. The other authors have declared that no competing interests exist.

7. Please provide an amended Funding Statement declaring the commercial affiliation ORTEC, as well as a statement regarding the Role of Funders in your study.

You can update author roles in the Author Contributions section of the online submission form.

“The funder provided support in the form of salaries for authors WJD and NS but did not play any additional role in study design, data collection and analysis, decision to publish, or preparation of the manuscript. The specific roles of these authors are detailed in the ‘author contributions’ section. ORTEC provided support in the form of the salary of author MeB. This author contributed to the development and updates of the app. He also reviewed the methods section, specifically the technical and developmental details of the app. He did not play any additional role in the data analysis or presentation of results in this manuscript. Maintenance of the app and salary of the researchers was funded by ZonMw (previously mentioned).

Within your Competing Interests Statement, please confirm that this commercial affiliation does not alter your adherence to all PLOS ONE policies on sharing data and materials by including the following statement: 

"This does not alter our adherence to PLOS ONE policies on sharing data and materials. The restrictions on sharing of data are due to national legal regulations.” 

 

Reactions to additional Editor Comments:

1. Reviewer #1: This is interesting work. I've a number of comments and suggestions that I think could further improve the quality of this manuscript; including better presentation, more acknowledgement of the potential limitations and biases present in a self-selecting app-based population, and some more formal analysis to support the claims being made.

Thank you for the acknowledgment of the relevance of this work. We are very grateful for the reviewer’s feedback. 

2. Abstract, line 61: should be have, not having. And I'm not entirely sure what you mean by "Symptom variables increased noticeably in the week" - do you mean reports of users experiencing symptoms increased?

We reviewed the text of the manuscript by a native English speaker. 

We changed this (line 55-57): Amongst users testing positive for SARS-CoV-2, the proportion of observations reporting symptoms was higher than that of the cohort as a whole in the week prior to a positive SARS-CoV-2 test.

3. Were there any limits on who could sign up to the app - e.g., were children allowed to? Were users encouraged to report on behalf of others on the multiple accounts, or were people asked to each fill out their own reports? 

The only limit on signing up for the app was access to a smartphone. Children were allowed to download and use the app. Users had the option of reporting on behalf of others. Over half of the observations are from individual users reporting only about themselves. 

We have added this to the manuscript (line 110): Children are allowed to download and use the app. Over 85% of the households with minors under 18 years of age are linked to an adult smartphone.

4. Line 123 - I'd prefer 'individual with Covid-19' to the more colloquial 'Corona patient'.

We added in line 128: “been exposed to an individual with COVID-19”

5. Giving user feedback in the app may alter their behaviour and make the app less representative of the population behaviours. Did you see any evidence of this? I think this merits a bit more discussion that already present as it has the potential to significantly bias your app population, and undermine the ability of the Covid RADAR to detect wider population changes. 

We agree that COVID radar app users’ behavior may not be representative of the population as a whole due to feedback from the app leading to an altering of behavior. The effect of this feedback loop on users’ behavior would be expected to lead to an overly conservative estimate of the behavior of the population. Still logical changes in reported behavior in the periods following national holidays and changes to social distancing policies are seen in COVID radar data. The possibility of this bias is acceptable given the goal of our study is to demonstrate patterns, rather than quantify associations. Additionally, altered behavior due to app-feedback would be expected to be observed in more loyal users of the app. Our sensitivity analysis on loyal users showed no significant difference reported behavior. Given these realities, while we accept that app feedback altering user behavior has the potential to bias our results, we feel any bias introduced has been shown here to be small. 

Added in discussion line 357-366: This may bias the generalizability of COVID radar data, especially behavioral data. The effect of this feedback loop on users’ behavior would be expected to lead to an overly conservative estimate of the behavior of the population. Despite this, expected changes in reported behavior in the periods following national holidays and changes to social distancing policies are observed in COVID radar data. Additionally, altered behavior due to app-feedback would be expected to be observed in more loyal users of the app. Our sensitivity analysis on loyal users showed no significant difference in reported behavior. Given these realities, while we accept that app feedback altering user behavior has the potential to bias our results, we feel any bias introduced has been shown here to be small. 

6. I think at times the authors have put too much into the supplementary material and it has prevented the manuscript from being self contained. For example, line 135 "Following the data cleaning process detailed in the online supplement..." It may be that the cleaning process is extensive and a full description of it best belongs in a supplement, but I think it would help readers if a brief overviewing of the process was included in the main text. 

We added a brief overview of the cleaning process of the data in the main text (line 134-139): Data were transferred daily to a safe data environment within the Information Technology system of the LUMC (supplemental Figure 2). Following importation of the daily data, we exclude observations from users who had requested to opt out, observations listing nonexistent postcodes, and double measurements within one user. We considered users COVID-19 positive/negative if they indicated a COVID-19 test result at least twice in the app, with the date of the first report used as day zero. 

7. Many of the figures don't have labels for the y-axis, please add. 

We have added Y-axis labels to all figures. 

8. There are a huge number of figures. It's good to have included so much detail, but I think some of these could be condensed into single figures with multiple panels. I think this would make it easier for readers to engage with and understand your work; it is difficult to do so when frequently scrolling between the texts and figures. For example, figures 5 and 6 could be combined into one.

We grouped all supplementary material, resulting in several panels with 7-11 figures each. Figure 5 and 6 are combined into one panel (figure 5 a and b), just as Figure 7 and 8 (figure 6 a and b). 

9. It is well documented that populations of participatory digital surveys tend to not be representative of the population - they tend to be more female, younger, more educated. It appears the Covid RADAR is no exception; it would be helpful if this was made explicit by comparison of your demographic to the country's general population, and the implications of this discussed. There are also harder to quantify biases: signing up to the app suggests a greater interest in Covid-19 than the average person, which might mean e.g. they are more likely to take the rules seriously and stick to them. It is very difficult to quantify the effect of this but it should be acknowledged as a limitation. 

The remarks of the reviewer are correct. We added a figure in the supplements (S.5) displaying the demographics of the Netherlands and the COVID radar app. The educational level of the users of the COVID radar app is unknown. The several sensitivity analyses we performed suggest a small bias towards more engagement to social distancing rules in more loyal users. We added this to the discussion.

Added to Results line 214-215: Female users were overrepresented compared to the national figures.

Added to Discussion lines 339- 342: COVID radar users were more often female and middle-aged. This was due to the overrepresentation of healthcare workers (who were more often female and middle-aged). However the sensitivity analysis excluding healthcare workers showed no different conclusions. 

10. It is surprising that the Covid positive population tended to be older than the negative population. Could you comment on this - does this finding agree with other data sources?

We assume the reviewer is commenting on table 1. In this table we compare users who report a positive COVID-19 test to users who don’t; (we do not report demographics from negative testing users!). The change of reporting a positive COVID-19 test is higher in more loyal app users, who are on average older. In addition, during the first two COVID-19 waves in the Netherlands, younger inhabitants (<12 years) often remained untested.

 Added to Discussion line 374-376: Also, testing of underaged (<12 years) was rare during the study period. Resulting in a relative old SARS-CoV-2 positive cohort in this study.

11. Figure 7: are you able to show number of tests conducted as well as number of positive tests; as it's difficult to see whether the peaks are true rises in population Covid-19 or just due to increased testing. The comparison is qualitative, and should be supported by measuring the quantiative correleation between your app data and cases; this would also help you formally justify your claim that the correlation is higher in regions with greather app participation.

During the research period testing policy indeed varied. The number of tests in December was almost twice as high as in October, but the number of positive tests were almost the same. However, we would emphasize that app participation was independent of formal test practices in The Netherlands. In addition, the number of users did not change during change in test policy. Therefore we believe it will not influence the observation that correlations between app reported cases and true rises in population COVID-19 in those regions with greater app policy is affected by change in test policy.

12. Your claims that "‘fever’, ‘pain in the chest’ and ‘loss of smell’ are associated with COVID-19 case count while variables ‘coughing’ and ‘sore throat’ correlated more

closely with Rhinovirus cultures" is again, completely based on qualitative comparison and would be strenghtened by quantitative measurements. These findings appear to me consistent with findings in the literature, which is encouraging, and I think this should be noted.

The analyses of symptoms specific to Rhinovirus or COVID virus are indeed qualitative due to the data structure and in our analysis there is no possibility to perform more quantitative analyses on this topic. However, inspired by your remark we have added references to prior research on symptomaticity of Rhinovirus and COVID-19, indeed consistent with our findings. 

See discussion lines 306: “These associations are consistent with those seen in prior research [13-16].”

Reviewer #2: The submitted manuscript reports analysis from data obtained from COVID RADAR app available in The Netherlands during COVID-19 pandemic outbreak (2020-2021). Overall, the analysis makes sense and the results are coherent. I have only a minor issue to address. As the app is based on self-reporting by app users, it should be specified that a bias toward serious illness exists, as the hospitalized users into Intensive Care Unit are likely not reporting their symptoms. 

This is a limitation in the analysis because it would be have been interesting to infer which symptoms and which areas might be prone to serious illness, i.e. also try to assess if pollution is playing a role. Other minor comments regard the manuscript, where sometimes too colloquial words are used, e.g. Corona. Lastly, on plots, x and y axis labels should be added.

We thank this reviewer for this feedback. We reviewed the manuscript for colloquial words and added labels to all axis. It is true that users who are admitted to the ICU are not able to report their symptoms. We did not mention the bias toward serious illness in the discussion because we believe this is outside the scope of this manuscript (which is ‘description of COVID radar data’) and outside the ultimate goal of this tool: to predict regional casecounts before positive COVID-19 tests are known. At the time patients are admitted to the ICU a result of a SARS-CoV-2 test is often known, given the deterioration of patients with COVID-19 is in the second week of symptomaticity (see Chen, et al. (2020). Pattern of Deterioration in Moderate Patients with COVID-19: A Multi-Center, Retrospective Cohort Study.). So this category of users is not relevant for the purpose of this study.

The remarks and suggestions for analyses of areas that might be prone to serious illness are not possible with COVID radar’s data because of no registration of individual data about ICU admission within the app, and flawed data about regional ICU admissions due to intensive inter ICU transportations of patients. However, analysis of regional differences of long term lasting COVID-19 symptomaticity (for example shortness of breath) and the regional distribution of them (enriched with data about air pollution) is possible with COVID radar’s data. This was outside this manuscript’s scope, but we have added this in suggestions for further research in the discussion (lines 386-388): Another possibility for further research is testing of associations between regional SARS-CoV-2 cases, symptoms, behavior and other regional data about for example related to the physical environment.

---

## [Decision Letter · Decision Letter 1]

9 Jun 2021

COVID RADAR app: Description and Validation of Population surveillance of Symptoms and Behavior in relation to COVID-19.

PONE-D-21-03787R1

Dear Dr. Kiefte,

We’re pleased to inform you that your manuscript has been judged scientifically suitable for publication and will be formally accepted for publication once it meets all outstanding technical requirements.

Kind regards,

Simone Lolli

Academic Editor

PLOS ONE

Additional Editor Comments (optional):

Reviewers' comments:

Reviewer's Responses to Questions

**Comments to the Author**

1. If the authors have adequately addressed your comments raised in a previous round of review and you feel that this manuscript is now acceptable for publication, you may indicate that here to bypass the “Comments to the Author” section, enter your conflict of interest statement in the “Confidential to Editor” section, and submit your "Accept" recommendation.

Reviewer #1: All comments have been addressed

Reviewer #2: All comments have been addressed

2. Is the manuscript technically sound, and do the data support the conclusions?

Reviewer #1: Yes

Reviewer #2: (No Response)

3. Has the statistical analysis been performed appropriately and rigorously? 

Reviewer #1: Yes

Reviewer #2: Yes

4. Have the authors made all data underlying the findings in their manuscript fully available?

Reviewer #1: Yes

Reviewer #2: Yes

5. Is the manuscript presented in an intelligible fashion and written in standard English?

Reviewer #1: Yes

Reviewer #2: Yes

6. Review Comments to the Author

Reviewer #1: (No Response)

Reviewer #2: I am happy that the authors addressed all my previously raised issues. Now the manuscript is ready for publication after technical corrections, e.g. typos, low-res

7. PLOS authors have the option to publish the peer review history of their article (what does this mean?). If published, this will include your full peer review and any attached files.

Reviewer #1: No

Reviewer #2: No

---

## [Editor Report · Acceptance letter]

21 Jun 2021

PONE-D-21-03787R1 

COVID RADAR app: Description and Validation of Population surveillance of Symptoms and Behavior in relation to COVID-19 

Dear Dr. Kiefte-de Jong:

I'm pleased to inform you that your manuscript has been deemed suitable for publication in PLOS ONE. Congratulations! Your manuscript is now with our production department. 

Kind regards, 

on behalf of

Dr. Simone Lolli 

Academic Editor

PLOS ONE